# Effect of Ethanol-Induced Methyl Donors Consumption on the State of Hypomethylation in Cervical Cancer

**DOI:** 10.3390/ijms24097729

**Published:** 2023-04-23

**Authors:** Xiao Han, Fang Fang, Weiwei Cui, Ya Liu, Yuxin Liu

**Affiliations:** Department of Nutrition and Food Safety, School of Public Health, Jilin University, 1163 Xin Min Street, Changchun 130021, China

**Keywords:** alcohol, cervical cancer, DNA methylation, epigenetics

## Abstract

Cervical cancer causes malignant tumors in females and threatens the physical and mental health of women. Current research shows that persistent infection of high-risk HPV is the main cause of cervical cancer. However, not all cervical cancer is caused by HPV infection, which may also be related to other factors, such as nutritional status and lifestyle. This study focuses on the effect of alcohol consumption on the methylation status of cervical cancer from the perspective of methyl donors. We established a mouse tumor-bearing model with cervical cancer SiHa cells, and at the same time, we cultured SiHa cells in vitro. Different concentrations of ethanol were administered to the model mice and SiHa cells. Then, we detected the levels of the methyl-donor folate and methionine and their metabolite homocysteine levels in mice serum, tumor tissues, and SiHa cells. Furthermore, we determined the expression of the members of the DNA methyltransferase family (DNMT1, DNMT3a, and DNMT3b) in tumor tissue by immunohistochemistry. qRT-PCR and Western blotting analysis were used to detect the mRNA and protein levels of members of the DNA methyltransferase family in cervical cancer SiHa cells. Our results show that the levels of the methyl donor (folate and methionine) decreased with the increase of ethanol concentration (*p* < 0.05), and the homocysteine level increased significantly (*p* < 0.05). In SiHa cells, the mRNA and protein levels of the DNMT family members and their receptors were significantly higher than those in the control group (*p* < 0.05). Collectively, these results suggest that ethanol could influence DNMT expression by inducing methyl donor consumption, thereby causing cervical cancer cells to exhibit genome-wide hypomethylation.

## 1. Introduction

Cervical cancer causes malignant tumors that seriously threaten women’s health. The 2018 Global Tumor Epidemiology Statistics (GLOBOCAN) report showed that among adult women aged 25–59 years, the incidence and mortality of cervical cancer ranks second among malignant tumors [1]. In 2020, there were 604,000 new cases of cervical cancer worldwide, and the total number of cases and deaths accounted for 6.5% and 7.7% among female tumors, respectively [2]. Current research results have shown that the main pathogenic factor of cervical cancer has been identified as continuous infection of high-risk human papillomavirus (HPV) [3]. In addition, dietary and lifestyle factors, such as smoking, drinking, age of first sexual intercourse, multiple sexual partners, and so on, are risk factors for cervical cancer development, recurrence, and mortality [4,5]. With the development of epigenetics, reports have shown that changes in DNA methylation levels and patterns are closely related to the occurrence and progression of tumors [6]. Current studies have shown that tumor tissues usually present genome-wide DNA hypomethylation [7]. DNA methylation requires enzymes and methyl donors. DNA methyltransferase adds methyl group cytosine residues in a cytosine-guanine (CG) pair to generate 5-methylcytosine, which results in changes in DNA conformation, stability, and interactions with proteins. This kind of abnormal methylation of the genome is critical for the occurrence of tumors [8]. The family of DNA methyltransferases mainly includes DNMT1, DNMT3a, and DNMT3b. These three members have different biological functions when maintaining methylation and remethylation. DNMT3a and DNMT3b are primarily responsible for de novo methylation during embryonic development, and DNMT1 is mainly responsible for maintaining the methylation level during cell division [9].

Methyl donors are substances that provide methyl groups during the metabolic process of the body and are necessary for the synthesis of many important substances. The main way the body obtains methyl groups is through food intake, such as by consuming folic acid, methionine, and choline. The intake of methyl donors affects the level of DNA methylation, which affects the epigenetics of cancers [10,11]. In addition, studies have also reported that DNA methylation may directly supplement HPV screening as a molecular diagnostic and prognostic marker for cervical intraepithelial neoplasia and cervical cancer [12,13].

Alcohol consumption is also an important risk factor for tumor development; according to WHO data, 3.5% of all cancer-related deaths are related to chronic alcohol drinking [14]. Alcohol can induce carcinogenesis in numerous organs, including the upper aerodigestive tract, liver, colorectum, female breast, etc. [15]. However, there are few reports on whether alcohol consumption is carcinogenic to cervical cancer [16,17]. Therefore, this study aimed to explore the influence of alcohol on the methyl donor status and the expression levels of DNA methyl transferase family proteins in cervical cancer in order to support the role of alcohol as a risk factor for cervical cancer carcinogenesis, and at the same time, to help elucidate the important role of epigenetics in the pathogenesis of cervical cancer. Since alcohol is a compound containing an R-OH group and other impurities, ethanol was selected for gradient dilution in our experiment to reduce confounding factors.

## 2. Results

### 2.1. The Effect of Ethanol on the Tumor-Bearing Mice Model and SiHa Cells

The weight of mice in each group was weighed every 3 days, and the tumor volume of mice in each group was measured. As shown in Figure 1A,B, with the increase of ethanol treatment concentration, the weight of mice was significantly lower than that of the Tum group. On the contrary, the tumor volume increased with the increase of the ethanol dose. After the animals were killed, the tumor tissue of each group was weighed. The results showed that the tumor weight in the high-dose ethanol group (THA group) was significantly higher than that in the control group (Tum group), and with the extension of ethanol treatment time, the tumor weight in the medium-dose and high-dose ethanol groups at 6 weeks was significantly higher than that at 4 weeks (see Figure 1C for details).

The CCK-8 method was used to detect the toxic effects of different concentrations of ethanol in SiHa cells, which were cultured for 24 h, 48 h, and 72 h. These results are shown in Figure 1D. With increasing ethanol concentrations, the survival rate of SiHa cells decreased significantly in a dose-dependent manner. The 24 h IC50 value was 515.281 mM. Therefore, in this study, an ethanol concentration close to the IC50 was selected as the medium dose, and the upper and lower doses were also included as follow-up doses: 0 mM, 200 mM, 400 mM, and 800 mM.

### 2.2. Effects of Ethanol on Methyl Donors Levels

After intervention with different concentrations of ethanol, the serum levels of folic acid in each group showed a significantly downward trend compared to the Tum group (at 4 w, F = 7.382, *p* < 0.001, at 6 w, F = 19.230, *p* < 0.001). Moreover, the serum folic acid levels in tumor-bearing mice treated with high doses of ethanol for 6 w were lower than those in the mice treated for 4 w (t = 2.953, *p* = 0.042). The results are shown in Figure 2A. In addition, the folic acid levels of tumor tissues in tumor-bearing mice treated with medium and high doses of ethanol decreased more significantly than in the Tum group at 4 w or 6 w, respectively (Figure 2B). Similarly, after 3 d of treatment with different concentrations of ethanol, the folic acid contents in SiHa cells in the medium- and high-dose groups were significantly lower than those in the control group of cells (F = 31.756, *p* < 0.001), while at 6 d, the intracellular folic acid content in only the high-dose group was significantly reduced (F = 8.844, *p* = 0.006). The results are shown in Figure 2C.

After 4 w and 6 w of low-dose, medium-dose, and high-dose ethanol interventions, the content of methionine in the serums of each group decreased significantly (at 4 w, F = 31.488, *p* < 0.001, at 6 w, F = 11.515, *p* < 0.001), and the results are shown in Figure 2D. In addition, the methionine levels in the tumor tissues of the animals treated with ethanol decreased more significantly than the Tum group at 4 w (F = 23.359, *p* < 0.001), and the levels in the medium- and high-dose groups were obviously lower than the Tum group at 6 w (F = 4.815, *p* = 0.034). The results are shown in Figure 2E. In SiHa cells, the levels of methionine in each group decreased significantly compared to the control group after ethanol intervention for 3 d, but no significant change was observed at 6 d in each group. (Figure 2F).

Homocysteine is generated after the methyl donors are depleted to reflect DNA methylation levels indirectly. The results are shown in Figure 2G, where it can be seen that the levels of homocysteine in the serums of the mice in each group increased notably (at 4 w, F = 26.739, *p* < 0.001, at 6 w, F = 57.283, *p* < 0.001). Moreover, the serum homocysteine level in tumor-bearing mice treated with high doses of ethanol for 6 w were higher than in those treated for 4 w (t = −3.832, *p* = 0.019). In addition, the homocysteine levels of the tumor tissues in tumor-bearing mice treated with ethanol for 4 w or 6 w increased more significantly, respectively (Figure 2H). When SiHa cervical cancer cells were treated with ethanol for 6 d, the level of extracellular homocysteine in the medium-dose group increased significantly (*p* < 0.05). The results are shown in Figure 2I.

### 2.3. The Effect of Ethanol on the DNA Methylation Level

This study detected the genomic DNA methylation level in cervical cancer tumor-bearing mice and SiHa cells after treatment with different concentrations of ethanol. The results are shown in Figure 3. In the tumor tissues of tumor-bearing mice, it was concluded that the DNA methylation levels in medium- and high-groups were lower than in the Tum group at 4 w, and the levels of DNA methylation in each ethanol intervention group decreased obviously at 6 w (*p* < 0.05). Moreover, the DNA methylation levels in each group at 6 w were lower than those at 4 w. The results are shown in Figure 4A. As Figure 4B shows, when SiHa cervical cancer cells were treated with ethanol for 6 d, the level of DNA methylation in each group decreased significantly (*p* < 0.05).

### 2.4. The Effect of Ethanol on the Expression of Methyltransferase Family Members in Tumor Tissues

In this study, tumor tissue sections were stained in situ by immunofluorescence. The results show that with an increasing dose of ethanol, the expression of the proteins in the DNMT family is significantly increased; these results are shown in Figure 4. More intuitively, we quantified the results of immunofluorescence, and the results are shown in Figure 5. The results showed that the expression level of DNMT1 was significantly higher than that of the Tum group when treated with ethanol for 4 w, and the expression level of DNMT1 in the high-dose ethanol group was significantly higher than that in the control group at 6 w. Moreover, the expression level of DNMT1 in the high-dose ethanol group treated with ethanol for 6 w was significantly higher than that in the group treated with ethanol for 4 w (Figure 5A). According to the DNMT3a expression level results, both medium and high doses of ethanol could increase the expression levels of DNMT3a at 4 w and 6 w compared with the Tum group, respectively (Figure 5B). Similarly, the expression levels of DNMT3b were higher in the low-, medium-, and high-dose groups than in the Tum group at 4 w and 6 w (Figure 5C). The expression levels of Mecp in low-, medium-, and high-dose groups were all higher than that in the Tum group after ethanol treatment for 4 w, and the expression level of Mecp in the low- and high-dose groups was significantly higher than that in the Tum group at 6 w (Figure 5D). It could be seen that the response of DNMT family members to ethanol treatment tends to increase, but the specific points of increase are different.

### 2.5. The Effect of Ethanol on the mRNA Expression of Methyltransferase Family Members in SiHa Cells

After treatment with low, medium, and high doses of ethanol for 3 and 6 d, the mRNA expression levels of the *DNMT* family of genes in SiHa cells increased significantly, and the difference was significant. The results are shown in Figure 6. The effect of ethanol on the mRNA expression level of *DNMT 1* was significantly higher in the low-, medium-, and high-dose groups than in the Tum group when treated with ethanol for 3 d and 6 d, respectively. Moreover, the expression level of *DNMT 1* in the high-dose group was significantly higher after ethanol treatment for 6 d than that for 3 d (Figure 6A). The effect of ethanol on the mRNA expression level of *DNMT 3a* was significantly higher in the low-, medium-, and high-dose groups than in the Tum group when treated with ethanol for 3 d and 6 d, respectively. However, the expression levels of *DNMT 3a* in the medium- and high-dose groups were significantly lower after ethanol treatment for 6 d than those for 3 d (Figure 6B). Similarly, the effect of ethanol on the mRNA expression level of *DNMT 3b* was significantly higher in the low-, medium-, and high-dose groups than in the Tum group when treated with ethanol for 3 d and 6 d, respectively. However, the expression levels of *DNMT 3b* in the low-, medium-, and high-dose groups were significantly lower after ethanol treatment for 6 d than that for 3 d (Figure 6C). This suggests that longer ethanol treatment may affect *DNMT 3a* and *DNMT 3b* activities due to increased methyl donor consumption. As for the mRNA expression level of *Mecp*, after ethanol treatment for 3 days, *Mecp* expression levels in all groups were higher than that of the Tum group, and after ethanol treatment for 6 days, *Mecp* expression levels in medium- and high-dose groups were significantly higher than those of the Tum group (Figure 6D).

### 2.6. The Effects of Ethanol on the Expression of Methyltransferase Family Proteins in SiHa Cells

After SiHa cells were given different concentrations of ethanol, the levels of the DNMT family of proteins, its receptor, and the corresponding trends were similar to the mRNA. However, these changes were not as significant as those of the DNMT family mRNA expression levels after ethanol treatment. These results are shown in Figure 7.

## 3. Discussion

This study detected the levels of methyl donors (folic acid and methionine) and homocysteine after stimulation with low, medium, and high doses of ethanol and discussed the effects of ethanol on the changes in methyl donor levels and effects on DNA methylation in cervical cancer.

As an essential nutrient in the growth process, folic acid is closely related to DNA methylation. Folic acid acts as a one-carbon coenzyme in DNA synthesis and repair and is directly involved in the transfer of DNA methyl groups. Methionine is one of the essential amino acids of the body, and it is also the precursor for the production of S-adenosylmethionine (SAM). SAM provides a carbon unit and is converted into homocysteine via DNMT. Changes in SAM could cause the changing in homocysteine and DNMT. When methionine is consumed as a methyl donor, homocysteine is produced and released from the cell. Homocysteine is generated after the methyl donors are depleted, so detection of homocysteine levels can indirectly reflect DNA methylation levels. Our results showed that administration of different doses of ethanol for different treatment times resulted in a decrease in the level of methyl donors in cervical cancer tissues and cells and an increase in the level of homocysteine. This result is similar to the findings in other research on folate acid and DNA methylation studies [18,19]. These results all indicate that folic acid is an important methyl donor involved in DNA methylation. Meanwhile, some clinical studies have shown that low folic acid content (including blood folic acid concentration, folic acid intake, and/or folic acid intake) is associated with increased risk of cardiovascular diseases, multiple cancers, neural tube defects, and tumors [20]. In particular, studies on folic acid and tumor risk have shown that folic acid may be a protective factor for tumor risk. For example, after heterogeneity studies were excluded, adequate intake of folic acid was associated with a 27% reduction in ovarian cancer risk [21], while high intake of folic acid was associated with an 11% reduction in endometrial cancer risk, showing a marginal negative correlation [22]. Moreover, studies have shown that folic acid, as an important methyl donor, affects the expression of DNMT in cervical cancer cells, which is associated with DNA global hypomethylation [23].

As we all known, DNA methylation is a chemical modification that plays an important role in the regulation of epigenetic gene expression. DNA methylation is mainly catalyzed by three enzymes, DNMT1, DNMT3A, and DNMT3B, that is, the DNA methyltransferases (DNMTs). Furthermore, we explored the effect of ethanol on the expression of DNMTs in tumor tissue and SiHa cells. The results showed that treatment with different doses of ethanol for different times can lead to a significant increase in the mRNA expression levels of the DNMT family members in the cervical cancer tissues and cells, indicating that intracellular DNA methylation is activated. The reason for the activation of DNA methyltransferase may be two-sides. On the one hand, the direct effect of ethanol can promote epigenetic changes in cervical cancer tissues and cells; on the other hand, ethanol reduces the level of the methyl donor by promoting the consumption of one carbon unit such as folate and methionine, thus promoting the activation of the DNA methyltransferase family. Similarly, increasing evidence indicates that alcohol may induce epigenetic alterations, which also could be important contributory factors to alcohol-induced carcinogenesis [24,25]. There is supporting evidence that ethanol affects DNMT family expression levels, as well as DNA methylation status [26,27].

In summary, we used cervical cancer cells and tumor-bearing mice models to explore the effect of ethanol on the changes in methyl donors and the expression levels of DNMT family members. We found that ethanol can affect the growth of tumors by altering the level of intracellular DNA methylation. That is, ethanol could deplete the levels of methyl-donors (folate acid and methionine) in SiHa cells or mouse tumor-bearing tissue and activate the mRNA expression levels of DNMT family members (DNMT 1, DNMT 3a, DNMT 3b, and Mecp), resulting in a hypomethylation status of cervical cancer (Figure 8). However, current studies have shown that the lack of methyl donors leads to abnormal methylation patterns, low DNA methylation in the genome as a whole, and hypermethylation of CpG islands of tumor suppressor genes. We did not elaborate on the effect of alcohol on CpG islands and whether alcohol affects the occurrence of cervical cancer. The causal relationship between the level of methyl donors in cancer cells and the activation of the DNMT family was not clarified in this study.

## 4. Materials and Methods

### 4.1. Reagents

DMEM (HyClone, Marlborough, MA, USA); trypsin (HyClone); PBS, fetal bovine serum (Kangyuan, Shenzhen, China); absolute ethanol (purity > 99.99%); CCK-8 assays kits; mouse folic acid ELISA kit (Jianglai Bio, Shanghai, China); mouse homocysteic ELISA kit (Jianglai Bio); TRIzol; SYBR Green; cell lysate; protease inhibitor; DNMT 3a primary antibody (DNMT1, DNMT 3a, DNMT 3b, Mecp, Abcam, Cambridge, UK); and goat anti-rabbit secondary antibody were used in this study.

### 4.2. Mice

Experimental athymic nude mice were purchased from Beijing HFK Bioscience Co., Ltd. 90 SPF-level female mice, 4–5 weeks old, 18 g ± 2 g, mature, healthy, and not pregnant, were reared within a chamber barrier system. All feed, bedding, water, air, and items required for the experiment were sterilized by high-temperature and high-pressure or UV surface disinfection; all operators entering the laboratory were strictly trained and wore isolation clothing. All experiments were approved by the Animal Research Ethics Committees of School of Public Health Jilin University (NO. 2021-06.27).

### 4.3. Cells

SiHa cells are charactered by HPV16-positive cervical squamous cell carcinoma; therefore, in this study, SiHa cell line was selected and purchased from Fuheng Biological Cell Bank. DMEM high-sugar culture medium with 10% fetal bovine serum and 1% penicillin was used, and the cells were placed in a 37 °C cell incubator under 5% CO_2_ for cell growth. All cells were cultured in a 100 mm petri dish and passed or inoculated when the cells covered 70–80% of the bottom of the petri dish. The cultured SiHa cells were used for subsequent tumor-bearing and cell experiments in mice.

### 4.4. Establishment of a Mouse Tumor-Bearing Model

After 1 week of adaptive feeding, SiHa cells were implanted subcutaneously (1 × 10^7^ cells per mouse, in 0.1 mL) into the dorsal side of the mouse near the right iliac fossa under sterile conditions. Then, we observed the survival of the animals and the growth of the tumors. When the tumor volumes were about 0.5 cm^3^, different concentrations of ethanol were used for gavage. At this point, mice successfully inoculated with tumors were randomly divided into 4 groups. A total of 120 mice were selected in this study, of which 85 mice were observed with a tumor. Therefore, the number of mice in each group was 21 in the Tum group, 21 in the TLA group, 21 in the TMA group, and 22 in the THA group. According to the chemical substance toxicity database, the LD50 of ethanol to mice is 3450 mg/kg·bw (https://www.drugfuture.com/toxic/q55-q574.html (accessed on 20 April 2021)). We used the dose closer to LD50, which was 3000 mg/kg·bw as the highest concentration (THA group) of intragastric dose; twofold dilution was used as the medium dose (TMA group, 1500 mg/kg·bw; and the low dose of ethanol (TLA group, 750 mg/kg·bw) was half of the dose of the medium group. The Tum group was the group of tumor-bearing mice given an equal volume of normal saline instead of ethanol as a blank control group.

Mice bearing cervical cancer tumors were given low, medium, and high doses of ethanol solution by gavage, and the body weights and tumor sizes were measured every 3 days. The tumor volume was calculated as follows: tumor volume (cm^3^) = length × width ^2^/2. After 4 w of gavage, half of the mice were sacrificed for follow-up experiments, and the other experimental animals continued to be treated with ethanol until 6 w. The treatment process of tumor-bearing mice is shown in Figure 9.

### 4.5. Cytotoxicity Test

The CCK-8 method was used to detect the toxic effects of different concentrations of ethanol to SiHa cells. Then, a dose–time-response curve was generated, and the IC50 value of ethanol on it was determined. SiHa cells were inoculated into 96-well plates with 8000 cells per well and diluted with absolute ethanol (the final concentrations were 1600 mM, 800 mM, 400 mM, 200 mM, 100 mM, 50 mM, 25 mM, 12.5 mM, 6.25 mM, and 0 mM). After culturing for 24 h, 48 h, and 72 h, the medium was replaced with DMEM containing 10% CCK-8 reagent, followed by culturing for 2 h. Next, a microplate reader was used to detect the OD value of each well at 450 nm. The rate of relative viability was determined with the following formula: cell survival rate = (experimental group OD value-blank OD value)/(control group OD value-blank OD value) × 100%. The survival curve was plotted after ethanol treatment.

### 4.6. Detection of Folate Content in Serum, Tumor Tissue of Mice, and Cells

Folic acid contents were determined using an enzyme-linked immunosorbent assay (ELISA) kit. The folic acid test sample was sequentially added to the coated microplate and then combined with an HRP-labeled detection antibody to form an antibody-antigen-enzyme-labeled antibody complex. After thorough washing, the substrate TMB was added to develop color. TMB produces a blue color after catalysis by the HRP enzyme and a yellow color in an acidic environment. The intensity of the color is positively correlated with the folic acid content in the sample. The absorbance (OD value) was measured with a microplate reader at a wavelength of 450 nm, and the folic acid content was calculated by drawing the standard curve. The pretreatment and processing steps of the samples were carried out according to the instructions of the Jianglai Bio-Folic Acid Enzyme-linked Immunoassay Kit.

### 4.7. Detection of Methionine Levels in Serum, Tumor Tissue, and Cells

The methionine contents were detected by high-performance liquid chromatography. The serum, transplanted tumor tissue of mice, and cultured cervical cancer cells were collected for preprocessing. The specific method was as follows: after each sample was concentrated and dried, an appropriate amount (approximately 0.5 g) was weighed. In a 50 mL hydrolysis tube, 20 mL of 2M HCl was added, placed in an electric blast drying box, and hydrolyzed at 110 °C for 22 h. After removal and cooling, the tube was transferred to a 25 mL colorimetric tube with a constant volume.

Fifty microliters of the sample was accurately placed in a 15 mL centrifuge tube, placed in a vacuum drying oven, dried at 60 °C for 2 h (dry all solvents), filled with nitrogen, and accurately added to 50 μL of derivatization reagent: ethanol: phenyl isothiocyanate: water: triethylamine = 7:1:1:1 (prepared for immediate use, filled with nitrogen during preparation). The sample was derivatized at room temperature for 30 min, added to 2 mL of mobile phase A, mixed well, and tested on the machine through a 0.45 μm organic membrane.

The detection method was as follows: chromatographic column: C18 Shiseido 4.6 mm × 250 mm × 5 μm; injection volume: 10 μL; column temperature: 40 °C; wavelength: 254 nm; mobile phase: A: 0.1 mol/L anhydrous sodium acetate + acetonitrile = 97 + 3. After the sample was mixed, the pH was adjusted to 6.5 (31.815 g sodium acetate + 3880 mL of water + 120 mL of acetonitrile; B: acetonitrile + water = 80 + 20).

### 4.8. Determination of Homocysteine Levels in Serum, Tumor Tissue, and Cells

The three kinds of cells were inoculated in a 6-well plate at 2 × 10^4^ cells per well. DMEM containing 10% FBS was used to culture the cells at 37 °C under 5% CO_2_. When the cells adhered to the wall and started to proliferate, different concentrations of ethanol were added, and the cells were cultured for 3 d or 6 d, after which the DMEM and cells were collected. High-performance liquid chromatography was used to detect the homocysteine content in the culture medium, and the number of cells in each well was standardized. The collection of culture medium and cells was performed according to the instructions of the Chromsystems HPLC Kit. The experiment was repeated 3 times, and the average values are reported.

### 4.9. Determination of the Genomic Methylation Levels in Tumor Tissues and Cells

Three kinds of cells were inoculated in 6-well plates. After the cells were fully attached and started to grow, they were treated with different concentrations of ethanol for 3 d and 6 d, and the genomic DNA in each group of cells was extracted with a GeneJET Genomic DNA purification kit. After purification, genome methylation was quantified. Quantification of the whole genome methylation in each group of cells was performed using the MethylFlash Methylated DNA Quantification Kit, and the cell pretreatment and processing steps were carried out according to the manual.

### 4.10. Immunofluorescence

After the tissue sections were dewaxed, the sections were placed in antigenic repair solution for antigenic repair and then closed with primary and secondary antibodies. After DAPI staining, the sections were sealed with neutral resin for fluorescence photography.

### 4.11. qRT-PCR

The mRNA expression of the DNA methyltransferase family members was detected by qRT-PCR. The cells were seeded in a 60 mm petri dish and treated with different concentrations of ethanol for 3 d or 6 d. TRIzol was added to extract the intracellular mRNA, which was reverse transcribed into cDNA by a one-step method, and SYBR dye was used for real-time fluorescence quantification. The temperature conditions were as follows: 94 °C, 30 s; 94 °C, 5 s; 60 °C, 30 s, cycle 40 times. The primer sequences for the DNA methyltransferase family members are shown in Table 1.

### 4.12. Western Blot

The three kinds of cells were inoculated in a 60 mm petri dish, treated with different concentrations of ethanol for 3 d or 6 d, placed on ice, and washed twice with PBS that had been precooled to 4 °C. Then, 1% protease inhibitor-containing RIPA buffer was added to extract total protein from the cells. SDS-PAGE was performed. After electrophoresis, the proteins were transferred to PVDF membranes, blocked with 5% skim milk powder at room temperature for 2 h, and incubated with different dilutions of the primary antibody at 4 °C overnight. The antibody was discarded, and the membranes were washed with TBST for 10 min. The goat anti-rabbit secondary antibody was added for 45 min at room temperature 3 times and then discarded, and the membranes were washed with TBST for 10 min a total of 3 times then developed and imaged with ECL. The bands were assessed with grey-scale analysis.

### 4.13. Statistical Analysis

All results are expressed as the $\bar{X}\pm S$. Image J was used to quantify the gray values of the western blot bands. SPSS 24.0 software was used for statistical analysis. Differences between groups were compared by one-way ANOVA and pairwise comparison. Differences between two time points were determined by independent samples *t* tests. *p* < 0.05 was considered statistically significant. GraphPad Prism 6.0 was used to plot the data.

## Figures and Tables

**Figure 1 ijms-24-07729-f001:**
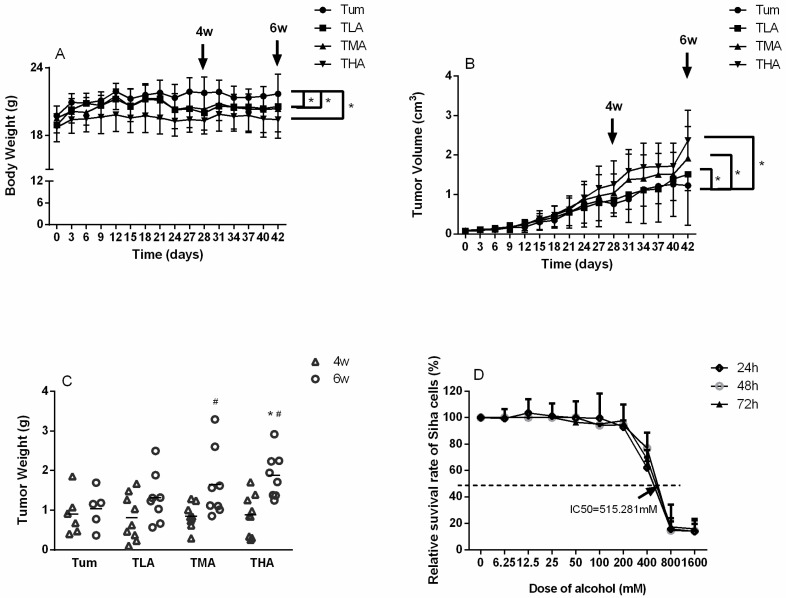
Distribution of body weights, tumor volumes, and tumor weights in cervical cancer-bearing mice. (**A**) The changing of body weight in each tumor-bearing mice group after treatment with low (TLA group), medium (TMA group), and high (THA group) doses of ethanol, Tum group is cancer-bearing mice without ethanol treatment. * *p* < 0.05, TLA, TMA, and THA groups compared with Tum group. (**B**) The changing of tumor volumes in each group, which were measured every 3 days. * *p* < 0.05, TLA, TMA, and THA groups compared with Tum group. (**C**) After the experimental animals were sacrificed, which were treated with ethanol at 4 w and 6 w, respectively, the weights of tumors were weighed. * *p* < 0.05, THA group vs. Tum group, # *p* < 0.05, 6 w vs. 4 w. (**D**) The relative survival rate of SiHa cells, which gave a 24 h IC50 value of 515.281 mM.

**Figure 2 ijms-24-07729-f002:**
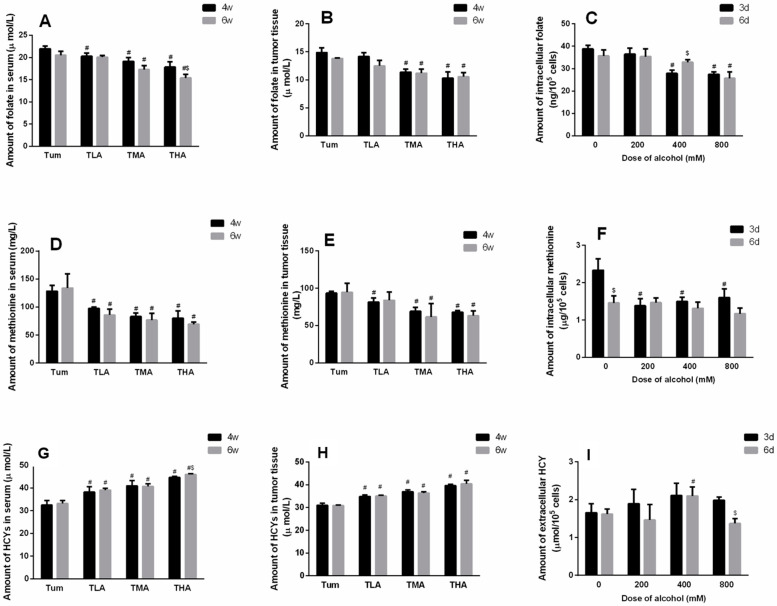
The effect of ethanol on methyl donor levels. (**A**) Folic acid levels in serum of tumor-bearing mice after treatment with low (TLA group), medium (TMA group) and high (THA group) doses of ethanol; Tum group is cancer-bearing mice without ethanol treatment. # *p* < 0.05, TLA, TMA, and THA groups compared with Tum group; $ *p* < 0.05 6 w vs. 4 w. *n* = 6. (**B**) Folic acid levels in the tumor tissues of tumor-bearing mice. # *p* < 0.05, TLA, TMA, and THA groups compared with Tum group. *n* = 6. (**C**) Folic acid levels in SiHa cells. # *p* < 0.05, 200 mM, 400 mM, and 800 mM compared with 0 mM. $ *p* < 0.05, 6 d vs. 3 d. *n* = 3. (**D**) Methionine levels in serum of tumor-bearing mice after treatment with low (TLA group), medium (TMA group), and high (THA group) doses of ethanol; Tum group is cancer-bearing mice without ethanol treatment. # *p* <0.05, TLA, TMA, and THA groups compared with Tum group. *n* = 6. (**E**) Methionine levels in the tumor tissues of tumor-bearing mice. # *p* < 0.05, TLA, TMA, and THA groups compared with Tum group. *n* = 6. (**F**) Methionine levels in SiHa cells. # *p* < 0.05, 200 mM, 400 mM, and 800 mM compared with 0 mM. $ *p* < 0.05, 6 d vs. 3 d. *n* = 3. (**G**) Homocysteine levels in serum of tumor-bearing mice after treatment with low (TLA group), medium (TMA group), and high (THA group) doses of ethanol; Tum group is cancer-bearing mice without ethanol treatment. # *p* < 0.05, TLA, TMA, and THA groups compared with Tum group, $ *p* < 0.05, 6 w vs. 4 w. *n* = 6. (**H**) Homocysteine levels in the tumor tissues of tumor-bearing mice. # *p* < 0.05, TLA, TMA, and THA groups compared with Tum group. *n* = 6. (**I**) Homocysteine levels in SiHa cells. # *p* < 0.05, 200 mM, 400 mM, and 800 mM compared with 0 mM. $ *p* < 0.05, 6 d vs. 3 d. *n* = 3.

**Figure 3 ijms-24-07729-f003:**
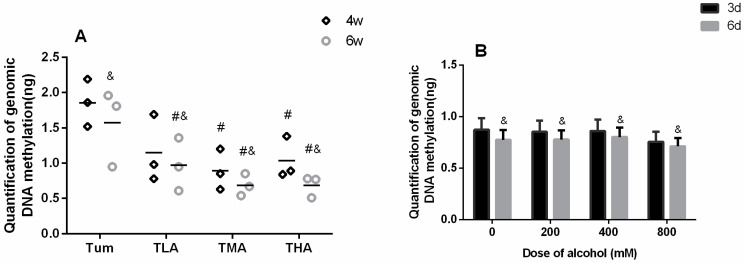
The effect of ethanol on the genomic DNA methylation level. (**A**) DNA methylation level in tumor tissue after treatment with low (TLA group), medium (TMA group), and high (THA group) doses of ethanol; Tum group is cancer-bearing mice without ethanol treatment. # *p* < 0.05, TLA, TMA, and THA groups compared with Tum group. & *p* < 0.05, 6 w vs. 4 w. *n* = 3. (**B**) DNA methylation level in SiHa cells. & *p* < 0.05, 6 d vs. 3 d. *n* = 3.

**Figure 4 ijms-24-07729-f004:**
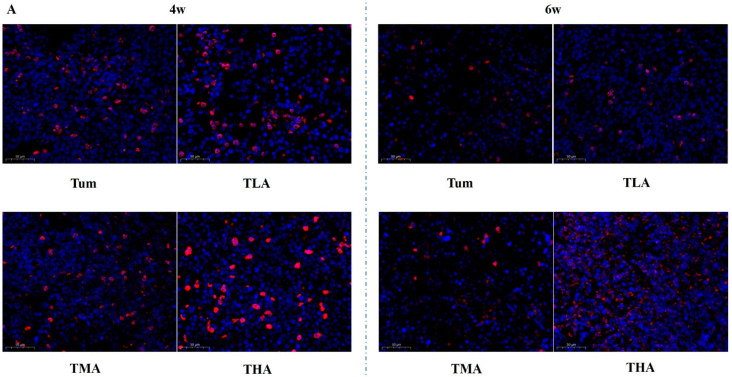
The effects of ethanol on the expression levels of DNMT family members in tumor tissues (magnification, 200×). (**A**) The expression levels of DNMT 1 after treatment with low (TLA group), medium (TMA group), and high (THA group) doses of ethanol in tumor-bearing mice groups, and the Tum group is tumor-bearing mice without ethanol treatment. (**B**) The expression levels of DNMT 3a. (**C**) The expression levels of DNMT 3b. (**D**) The expression levels of Mecp.

**Figure 5 ijms-24-07729-f005:**
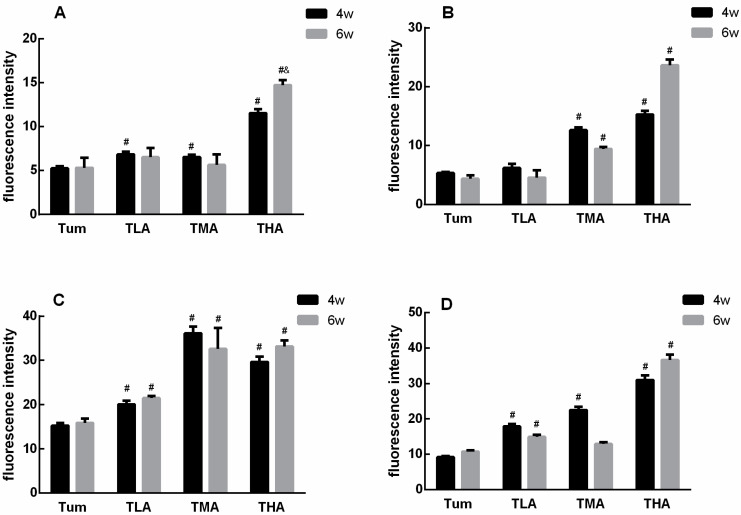
Statistical results of the DNMT family members fluorescence in each group. (**A**) The expression levels of DNMT 1 after treatment with low (TLA group), medium (TMA group), and high (THA group) doses of ethanol in tumor-bearing mice groups, and the Tum group is tumor-bearing mice without ethanol treatment. (**B**) The expression levels of DNMT 3a. (**C**) The expression levels of DNMT 3b. (**D**) The expression levels of Mecp. # *p* < 0.05, TLA, TMA, and THA groups compared with Tum group. & *p* < 0.05, 6 w vs. 4 w. *n* = 3.

**Figure 6 ijms-24-07729-f006:**
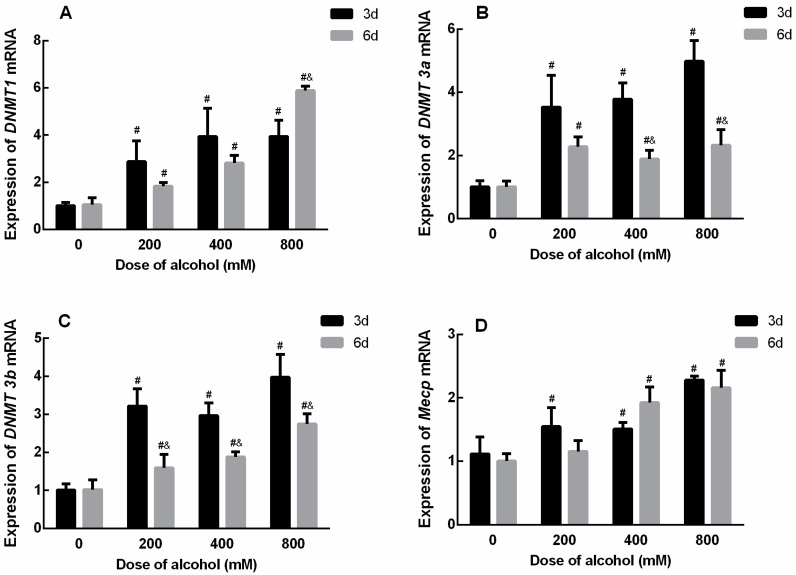
The effects of ethanol on the mRNA expression levels of *DNMT* family members in SiHa cells. (**A**) The mRNA expression levels of *DNMT1* (3 d, F = 8.516, *p* = 0.007; 6 d, F = 211.441, *p* < 0.001). (**B**) The mRNA expression levels of *DNMT3a* (3 d, F = 19.201, *p* = 0.001; 6 d, F = 9.870, *p* = 0.005). (**C**) The mRNA expression levels of *DNMT3b* (3 d, F =17.884, *p* = 0.001; 6 d, F = 22.538, *p* < 0.001). (**D**) The mRNA expression levels of *Mecp* (3 d, F = 16.365, *p* = 0.001; 6 d, F = 21.589, *p* < 0.001). # *p* < 0.05 vs. 0 mM. & *p* < 0.05, 6 d vs. 3 d. *n* = 3.

**Figure 7 ijms-24-07729-f007:**
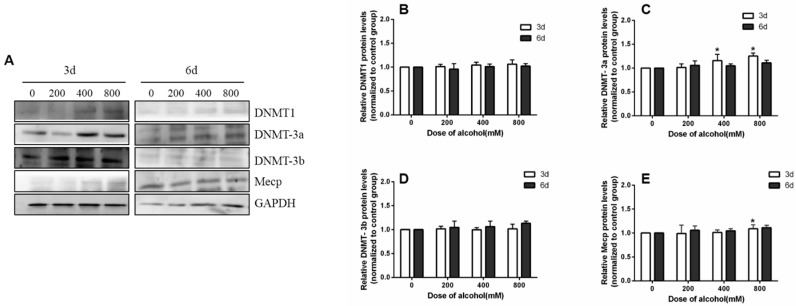
The effects of ethanol on the expression of DNMT family proteins. (**A**) The expression of DNMT family proteins in SiHa cells. (**B**–**E**) Protein quantification by image J software. * *p* < 0.05, 200 mM, 400 mM, and 800 mM compared with 0 mM.

**Figure 8 ijms-24-07729-f008:**
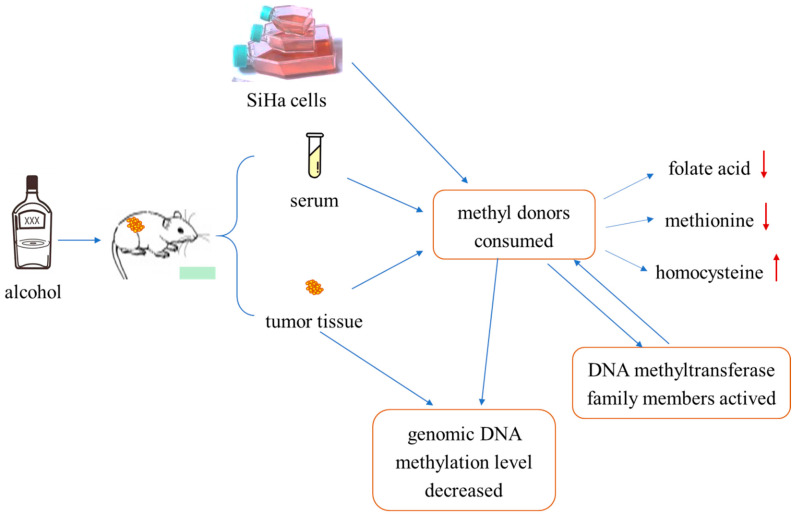
Summary of experimental procedure. Treating tumor-bearing mice or SiHa cells with ethanol resulted in decreased methyl-donor consumption in serum, tissue, and SiHa cells, that is, decreased folate and methionine levels and increased homocysteine levels. That process could activate members of the DNA methyltransferase family, thus causing genomic DNA hypomethylation in cervical cancer tissues and cells. In summary, ethanol can decrease the level of genomic DNA methylation through consumption of methyl donors or directly, and the consumption of methyl donors can interact with the activation of DNMA family proteins.

**Figure 9 ijms-24-07729-f009:**
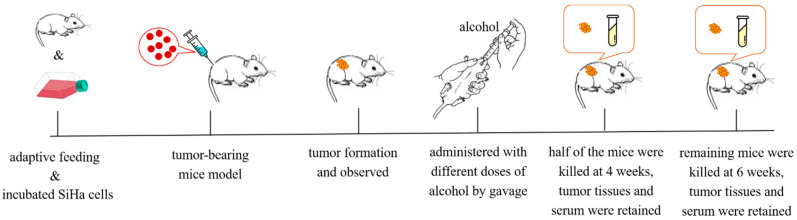
The treatment process of tumor-bearing mice.

**Table 1 ijms-24-07729-t001:** Sequences of qRT-PCR primers.

Gene Name		Sequences of qRT-PCR
DNMT 1	Forwaud	5′-agccgagcgagccagagatag-3′
	Reverse	5′-gagatgcctgcttggtggaatcc-3′
DNMT 3a	Forwaud	5′-cccaccagcataccctgagagtc-3′
	Reverse	5′-agacctttagccacgacccagac-3′
DNMT 3b	Forwaud	5′-agcagccctggagactcattgg-3′
	Reverse	5′-ctggttgcgtgttgttgggtttg-3′
Mecp	Forwaud	5′-tgctctgctcgcctcggatc-3′
	Reverse	5′-acctgcttccttctgcctcctg-3′
β-actin	Forwaud	5′-cagggcgtgatggtgggca-3′
	Reverse	5′-caaacatcatctgggtcatcttctc-3′

## Data Availability

The data presented in this study are available on request from the corresponding author.

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
