# Peer review of "Effect of Ethanol-Induced Methyl Donors Consumption on the State of Hypomethylation in Cervical Cancer"

_ijms, 2023, doi:10.3390/ijms24097729_

Round 1
Reviewer 1 Report (Previous Reviewer 3)
- Abstract and introduction sections are now much improved.
- Some grammatical inconsistencies are still present throughout the manuscript.
- “DNA methyltransferaseadds a methyl group cytosine residues in a cytosine-guanine (CG)…” missed a spacing between “methyltransferase” and “adds”.
- The results should be clarified that Tum group, as the control, is tumor without alcohol, whereas the intervention groups TLA, TMA, and THA, are still with tumor growth + varying levels of alcohol. As it reads currently, it can be misunderstood that Tum group has tumor growth, whereas the intervention groups do not (but alcohol is varied).
- For Fig.1, the panels A, B, C, and D are absent. Please add them to the figure. The legend for panel A shows statistical significance “* P < 0.05 vs. Tum group”, however the visualisation of these show comparisons to the THA group, so two of them are incorrect (only the THA vs Tum annotation is correct). For panel C, the note on statistical significance should be clarified “Tum group, # P<0.05 vs.6w” to ‘4w vs 6w’.
- The statement made for Fig.1 in text is “tumor weight in the high-dose alcohol group (THA group) was significantly higher than that in the control group (Tum group)”, however panel B shows statistical significance for all intervention groups, TLA, TMA, and THA group. This needs to be clarified.
- Discussion section reads better now.
- Fig. 8 title should be changed to be more descriptive. Perhaps something akin to “Summary of experimental procedure” or “proposed model…“ etc. I am unclear what the current title means.
Author Response
Please see the attachment.

Reviewer 2 Report (New Reviewer)
Cervical cancer is one of the most serious malignancies that threatens women's health. The etiology of cervical cancer is persistent, high-risk human papillomavirus infection. However, there are still other risk factors, such as alcohol consumption and other harmful lifestyle factors mentioned by the authors. In this paper, the authors studied the effects of alcohol consumption on the methylation level of cervical cancer through both animal and cell experiments. However, some questions need to be clarified and addressed.
1. There are many kinds of cervical cancer cell lines. Why were siha cells selected for the experiment on tumor-bearing mouse?
2. Figure 1 consists of 4 charts. Whether consider labelling them with A,B,C,D to make the results more clear? In addition, the bar of standard deviation should be added to the charts for body weight and tumor volume.
3. What is the basis for the treatment dosage of alcohol, Could you please attach the corresponding reference or website?
4. In discussion, I suggested to supplement the research of effect on methylation to the occurrence and progression in cervical cancer, which could enhance the depth and breadth of the discussion.
5. In materials and methods, the description of alcohol dosage should be ‘mg/kg·bw’, and the statistical method adopts the form of , instead of mean±SEM.
Author Response
Please see the attachment.

Reviewer 3 Report (New Reviewer)
As far as I am concerned, statistical analysis plays a pivotal role in this paper, and the overall analysis should be improved. The authors show some graphs that I find difficult to understand without having a proper statistical analysis.
The article also should point out what alcohol means. Evidently, it is Ethanol (CH3-CH2-OH), but it is only mentioned in the material section. It should be noted that the use of the word alcohol is only a simplification since alcohol is any organic compound that carries at least one hydroxyl functional group R-OH
Round 2
Reviewer 1 Report (Previous Reviewer 3)
All prior concerns have been addressed and the article is suitable for publication.
Reviewer 3 Report (New Reviewer)
As far as I am concerned I have no other comments
This manuscript is a resubmission of an earlier submission. The following is a list of the peer review reports and author responses from that submission.
Round 1
Reviewer 1 Report
1. What is the main question addressed by the research? The effect of alcohol-induced methyl donors consumption on the hypomethylation state of cervical cancer.
2. Do you consider the topic original or relevant in the field? Does it
address a specific gap in the field? Relevant in the field. Yes.
3. What does it add to the subject area compared with other published
material? Provides scientific evidence.
4. What specific improvements should the authors consider regarding the
methodology? What further controls should be considered? No.
5. Are the conclusions consistent with the evidence and arguments presented
and do they address the main question posed? Yes.
6. Are the references appropriate? Yes.
7. Please include any additional comments on the tables and figures. No additional comments.

Author Response
Response to Reviewer 1 Comments
- What is the main question addressed by the research? The effect of alcohol-induced methyl donors consumption on the hypomethylation state of cervical cancer.
Response 1: Yes, thank you for your affirmation. This study focuses on the effect of alcohol on the methyl donors (folic acid and methionine) in cervical cancer tissues and cells, resulting in a hypomethylated state of cervical cancer.
2. Do you consider the topic original or relevant in the field? Does it address a specific gap in the field? Relevant in the field. Yes.
Response 2: Thank you for your review. This study is not an original article in the field of cervical cancer methylation, but it may provide some support for the field. In particular, the effect of alcohol on the status of methyl donors in cervical cancer tissues and cells.
What does it add to the subject area compared with other published material? Provides scientific evidence.
Response 3: Yes, this study can provide a scientific basis for research on the effect of alcohol on the hypomethylation status of cervical cancer.
What specific improvements should the authors consider regarding the methodology? What further controls should be considered? No.
Response 4: Thanks for your recognition, we have further modified some contents in the article with unclear marks or language expression.
Are the conclusions consistent with the evidence and arguments presented and do they address the main question posed? Yes.
Response 5: Yes, the argument presented in this study is that alcohol could consume the methyl donors of cervical cancer tissues and cells, thereby leading to a hypomethylation state of cervical cancer. In addition, ELISA, qRT-PCR, immunofluorescence, Western Blot and other methods were used to detect relevant indicators in tumor-bearing mice and SiHa cells, thus proving the thesis of this study.
- Are the references appropriate? Yes.
Response 6: Thank you for your support, but we still add some references to better support the relevant results of this study.
Please include any additional comments on the tables and figures. No additional comments.
Response 7: Thanks for your comments, the author has modified the chart of the article in more detail to make it more clearly marked.
Reviewer 2 Report
Title: The effect of alcohol-induced methyl donors consumption on the hypo-methylation state of cervical cancer
Comments:
1) Why SiHa cell line was used? Why not any other cervical cancer cell lines?
2) Dose authors think if they use HeLa or C33A, they would expect the same result, and why?
3) How HPV status affects methylation and demethylation in addition to alcohol?
4) Why only one cell line was used? How would it affect if we want to make a conclusion in real life? Dose these results correlate with alcohol intake in cervical cancer cases?
5)The discussion section can be improved by correlating previous studies and what new information this work presents.
6) Dose the authors expect similar results, irrespective of squamous cell carcinoma or adenocarcinoma?
7) The significance of the study and the hypothesis needs to be more clearly stated in the introduction.
8) Authors have not given an explicit knowledge of previous reports and what novel information the current manuscript provides.
9) Also, a section on statistics and sample size power calculation be helpful.
10) Please check the typo and errors; for example, in the abstract, the ‘SiHa’ word has used inconsistency. How much variability can a system or user create on the same data?
Author Response
Response to Reviewer 2 Comments
1) Why SiHa cell line was used? Why not any other cervical cancer cell lines?
Response 1: Thank you for your question. Siha cells are derived from human cervical squamous cell carcinoma line, and at present, more than 70% of the clinical incidence of cervical cancer is squamous cell carcinoma, so SiHa cell line was selected in this study. At the same time, our team also conducted experiments on other cervical cancer cells such as Hela, C33A and Caski cells, which will be considered for further publication in another article.
2) Dose authors think if they use HeLa or C33A, they would expect the same result, and why?
Response 2: Yes, our team also conducted relevant experiments using Hela and C33A cell lines, and the results obtained were similar to those obtained by Siha cells. The reason may be that alcohol intake will consume the methyl donor in the cervical cancer cells and disturb the methylation state, thus leading to the hypomethylation in cervical cancer cells, which is independent of HPV infection and clinical typing of cervical cancer. These experimental results will be submitted in future articles.
3) How HPV status affects methylation and demethylation in addition to alcohol?
Response 3: Thank you for your question. Although this study only reported the research results of Siha cells, which are derived from human cervical squamous cell, HPV16+, our team conducted experiments on other types of cervical cancer cells. For example, Hela cells (HPV18+), Caski cells (HPV16 and 18+) and C33A cells (HPV-), the similar experimental results were obtained. All of which showed that alcohol could consume the methyl donor in cervical cancer cells, thus inducing the hypomethylation state of cervical cancer cells. Thus, HPV may have less effect on methyl donors than alcohol.
4) Why only one cell line was used? How would it affect if we want to make a conclusion in real life? Dose these results correlate with alcohol intake in cervical cancer cases?
Response 4: Thank you for your question. The Siha cells used in this study are derived from human cervical squamous cell carcinoma line, and the clinical incidence of cervical squamous cell carcinoma accounts for more than 70%. The selection of alcohol doses in this study were combined with the daily limit of alcohol recommended by dietary guidelines and dose conversion between human and mice, so as to support the risk factors of cervical cancer in reality. The results suggest that prolonged alcohol consumption in excess of the recommended daily amount may have an effect on the methylation status of cervical cancer.
5)The discussion section can be improved by correlating previous studies and what new information this work presents.
Response 5: Thanks for your suggestion, the author has further modified the discussion section to make it more consistent with the experimental results obtained.
6) Dose the authors expect similar results, irrespective of squamous cell carcinoma or adenocarcinoma?
Response 6: Thank you for your question. In addition to Siha cells (squamous cell carcinoma), our team also selected Hela cells (adenocarcinoma) for the experiment. The results obtained were similar to those obtained by SiHa cells, which proved that alcohol could consume the methyl donors (folic acid and methionine) in cervical cancer cells, thus affecting the hypomethylation state of cervical cancer. Therefore, both squamous cell carcinoma and adenocarcinoma may be affected by alcohol intake.
7) The significance of the study and the hypothesis needs to be more clearly stated in the introduction.
Response 7: Thanks for your suggestion, we have made further changes to the introduction of the manuscript.
8) Authors have not given an explicit knowledge of previous reports and what novel information the current manuscript provides.
Response 8: Thanks for your review. The author further modified the research progress on cervical cancer in the introduction, and provided evidence support for the incidence and influence of alcohol on cervical cancer through the experimental data of this study.
9) Also, a section on statistics and sample size power calculation be helpful.
Response 9: Thanks for your suggestion, we have improved the relevant content about statistical methods and sample size in the materials and methods.
10) Please check the typo and errors; for example, in the abstract, the ‘SiHa’ word has used inconsistency. How much variability can a system or user create on the same data?
Response 10: Thanks for your prompt, we have made further changes to the grammar, presentation and chart annotations in the article.
Reviewer 3 Report
This study has attempted to analyse the effect of alcohol consumption on various molecular markers and mechanisms in cervical cancer cells using a host mouse model, as well as cell cultures. In particular, the markers studied were related to methylation mechanisms, including folate, methionine, homocysteine, and DNMT. However, the methodology of these experiments is not fully transparent, and the results and discussion sections are lacking depth to properly convey the potential results found herein. Further to this, there is a lack of strength of statistical findings in corroborating the hypothesis, which warrants more cautious interpretation and wording of the findings. Overall, the results reported herein are not convincing, a problem which is compounded by the lack of explanation in the results and discussion section, with some lack of information in explaining methodology. There is some lack of transparency in the reporting of data, and the figures reported in the study suffer from poor legends and lack of (sometimes even absence of) in-text elaboration. The discussion section is also rather superficial, and seems to not address various key issues, relying instead on reiterating a pre-conceived idea that is supported by select data from the study (which cannot be fully ascertained due to the transparency issue mentioned above). This may be due to some language issues, from which the report suffers, but could also reflect potential reporting and experimental design biases that are not accounted for. The lack of referencing throughout the article also weakens the case presented. Ultimately, the study could benefit from a more rigorous presentation of results, as well as engaging in proofreading to ensure a clear narrative and minimisation of the abundant phrasing and grammatical issues throughout. I cannot, with a clear conscience, recommend the article for publication without major changes to almost every aspect of the article.
Abstract
- This statement in the abstract should be changed or removed: “However, the etiology and pathogenesis of cervical cancer are not very clear” as the pathogenesis of the vast majority of cervical cancer cases is well-linked to HPV infections. It may be true to say that those of unknown aetiology are poorly understood, but this sentence does not reflect this. Furthermore, this sets up the subsequent premise of the study, which is to investigate the effect of alcohol on methylation states in cervical cancer cells, perhaps better describing the latter subset (minority) of non-HPV cervical cancers.
- “In SiHa cells, the mRNA and protein levels of the DNMT family members and their receptors in the cells were significantly higher than those of the control group.” This statistical statement should be accompanied by the relevant threshold e.g. p<0.05.
- This statement seems very verbose and incorrect: “alcohol intervention can lead to the consumption of methyl-donor in cervical cancer cells and increase the expression levels of the DNA methyltransferase family members”, in particular regarding ‘alcohol intervention leading to consumption of …’ Perhaps paraphrasing would help better convey the conclusion regarding the relationship found between alcohol levels and expression levels of DNMT.
Introduction
- Similar to the comment in the abstract section, this statement doesn’t seem accurate: “the pathogenesis of cervical cancer is not very clear, and the main pathogenic factor of that is human papillomavirus (HPV) infection.” It may be true that we do not fully understand ALL the contributing factors to pathogenesis of cervical cancers, of which methylation states could be a major risk factor, but by far, HPV accounts for the highest risk amongst known factors, so it is disingenuous to suggest that the pathogenesis of cervical cancers, as a whole, is not very clear. As mentioned in the subsequent sentence, there are many other risk factors for the disease IN ADDITION to the main risk factor, HPV infection. To put this into context, perhaps mention of the relative risk contributions (percentagewise) would help readers.
- This statement, “the effect of DNA methyltransferase, a methyl group is added to the cytosine residues in a cytosine-guanine (CG) pair generating 5-methylcytosine, resulting in changes in DNA conformation, stability and interactions with proteins” merely addresses one type of DNA methylation, CpG methylation. Granted, this is the most common type of DNA methylation in humans, but as a general statement here, it is limited to only discuss CpG methylations, and incorrect to generalise that all methylation activity is of CpG methylations. Paraphrase here or include a more general statement about methylation. If the study here concentrates only on CpG methylation, it should be noted that this is the case, and why this choice was made.
- Inaccuracy in phrasing here, as ‘methyl’ is not a stand-alone substance, but a chemical side group (so it might be appropriate to paraphrase to ‘methyl-group’: “The main way for the body to obtain methyl is through food intake.”
- Could the authors provide clarification for this statement: “In addition, studies have also reported that DNA methylation may directly supplement or replace HPV screening as a molecular diagnostic and prognostic marker for cervical intraepithelial neoplasia and cervical cancer[12,13]”, as the two references here do not indicate, or even suggest that methylation analysis could replace HPV screening, but does have the potential to supplement such screening (unless I am misreading them). If indeed, this is not what the references have mentioned, then removing this statement is essential as to not mislead the reader regarding the significance of findings here (of replacement, though the supplementary role of methylation screening can be kept).
Results
- The first paragraph, including Fig.1 would be better suited in the methods section than in results section: “Mice with successful tumor bearing were given low, medium and high doses of alcohol solution by gavage, and the body weight and tumor tissue size were measured every 3 days. The tumor volume was calculated as follows: tumor volume (cm3 ) = length × width^2 /2. After gavage for 4w, half of the mice were sacrificed for follow-up experiments, and the other experimental animals continued to be treated with alcohol until 6w. The treatment process of tumor-bearing mice is shown in Figure 1.”
- Fig.2 requires a lot more clarification. Legends and figure text should include the abbreviations used (TLA = low alcohol treatment, TMA = medium alcohol treatment, THA = high alcohol treatment. What is Tum?) For each of the panels, e.g. Fig.2A, are the apparent differences between groups statistically significant? If so this should be indicated in the figure with appropriate labels, e.g. an asterisk, and the appropriate p-value of significance mentioned. If they are not statistically significant, this should be made clear and the result interpretations modified accordingly. This statement does not belong in the figure legend, rather in the methods section: “At 4 weeks, half of mice in each group were sacrificed for subsequent detection, and the remaining experimental animals were treated with alcohol until to 6 weeks.” It is sufficient to mark the time periods of 4w and 6w sacrifice, with the addition to the figure of a legend explaining the abbreviations 4w and 6w. Similarly, for panel C, statistically significant results should be labelled. I assume the symbols for $ and #$ are used to signify some measure of statistical significance, but it is not made clear in the legend, so these results cannot be overtly interpreted. Again, the description for panel C belongs better in the methods section: “After the experimental animals were sacrificed, the weight of tumors was weighed”; instead, consider using a description of the RESULTS herein.
- This sentence is not a result of the study and is better suited for the introduction or discussion section: “Methionine is one of the essential amino acids of the body, and it is also the precursor for the production of S-adenosylmethionine (SAM). SAM provides a carbon unit and is 4 converted into homocysteine via DNMT. SAM changes could cause the changing in homocysteine and DMNT.”
- This statement in-text does not match the data in Fig.3: “… folic acid levels in tumor-bearing mice treated with medium and high doses of alcohol decreased more significantly.” In Fig.3(A), only the THA group is marked as showing a significant difference between 4w and 6w, but the TMA groups isn’t marked as such. There may be a trend, as the authors suggest, but a dose-dependence cannot be established by comparison between two data points (either against the Tum sample, or between 4w and 6w); instead, a linear regression to show dose-dependence would be more appropriate. Similar argument for Fig.3(B) and (C). Furthermore, in panel C, the TMA data shows a significant increase in folate levels from 4w to 6w, which doesn’t corroborate the trend in other groups, as well as the progression from TLA to TMA to THA. This inconsistency puts into question the conclusion made (without context): “Meanwhile, the folic acid levels of tumor-bearing mice treated with high doses of alcohol in 6 w were more decreasing than 4 w (t=2.953, P=0.042)”, which is an untrue statement both statistically (no $ marker is given as per the legend), and the authors have conveniently omitted the statistically significant INCREASE in folate levels for the TMA group from 4w to 6w, of which, the 6w data is not significantly different from 0mM exposure to alcohol. Similar inconsistencies are observed in other panels, in particular, panel (I), where there is no apparent dose-dependent relationship between alcohol exposure and extracellular HCY. Indeed, it seems that the 6d levels of HCY are, where statistically demonstrable at 800mM alcohol, LOWER than at 3d, with a possible similar trend at other alcohol doses, though the latter are not statistically confirmed or denied. This observation warrants some addressing in the discussion section.
- Legend for Fig.3 is insufficient – all abbreviations should be elaborated, even if they seem to be apparent, e.g. Tum, HCY, TLA, TMA, THA etc.
- “Current studies have shown that tumor tissues exhibit low levels of genome-wide methylation.” Be this as it may, this is not a generalised statement on the findings herein, and should be moved to the introduction section with appropriate referencing.
- Fig.4 seems to only provide observable data in panel A, and no statistical information (or even non-statistical trend) is observable in panel B – perhaps the latter is worth relegating to an appendix. Furthermore, the comprehensiveness of the presented data is questionable – see comments on the methods section below.
- Fig.5 has potential to demonstrate some qualitative data, however there is neither enough information in the figures and legends, nor is there sufficient explanation in-text. Some quantifiable measure of measuring the expression levels, with annotations on significant changes in the panels is required to prove conclusively any changes. Further to this, the authors should provide a description on sampling of these visual fields – how was it ensured that these are random samplings representative of a homogenous observation, instead of fields specifically chosen to demonstrate a perceived difference (observation bias)? The figure legend should include identification of immunolabelling – what markers are identified with blue / red staining?
- Overall, the text description of the results section is insufficient, and merely references the figures with the expectation that the reader will find them self-explanatory (which they are not, especially without comprehensive legends).
- Fig. 6 is an archetype of the above critique – it is not self-evident what the expected and observed trends are here, especially since the statistically significant changes are labelled, but the trends are not addressed. A more comprehensive in-text description on why there is a dose dependence of alcohol and observed expression levels of DMNT / Mecp mRNA but a reversal of the trend between 3 and 6 days is essential for interpretation of these figures.
- This statement is confusing: “After various cervical cancer cells were given different concentrations of alcohol…” Was more than one cervical cancer cell-line experimented upon (SiHa)? If so, the details of this were not provided. If not, this sentence requires paraphrasing. The continuation of the sentence is equally confusing: “… protein levels and trends of the DNMT family and its receptor were close to the mRNA levels and trends” does this mean that the trends were similar? Perhaps a better choice of words is required, because it is not apparent what values are ‘close’ between the two sets of experiments, and even if some values were, the different nature of the experiments and biological interpretations would render this interpretation meaningless. Again, there is lack of clarity in the explanation of results in-text, and the legend for Fig.7 is not very helpful in elucidating the situation. The mention of J software is made briefly in the figure legend, but neither interpretation nor methodology is given in the appropriate sections.
Discussion
- The introduction, results, and discussion sections should be made clear that the conclusions drawn are based on in-vitro cell line analysis, as well as mouse tumour hosting (and that these are not in-vivo samplings). Furthermore, sample size of each experiment should be clearly mentioned (simply n=3 in parentheses, or equivalent, where results are mentioned).
- Neither references 14 nor 15 are contextually supportive of the statement in which they are referenced – they are not tumour methylation studies.
- True as this statement may be, there is no logical coherence in placing this sentence where it is: “Meanwhile, some clinical studies have shown that low folic acid content (including blood folic acid concentration, folic acid intake, and/or folic acid intake) is associated with increased risk of cardiovascular diseases, multiple cancers, neural tube defects, and tumors [16]”.
- This statement would serve better in an introduction section, as it does not discuss findings herein: “Alcohol consumption is also an important risk factor for tumor development, according to WHO data 3.5% of all cancer-related deaths are related to chronic alcohol drinking [19]. Alcohol could induce carcinogenesis in numerous organs, including the upper aero[1]digestive tract, liver, colorectum, and female breast, et al”. Further to this, correlation here (which are indeed risk factors) does not show causation, so this should not be used as a justification to validate correlations found in this study (which, themselves are not strongly statistically supported). The term ‘et al’ here is misused – perhaps the authors mean etc?
- In this statement: “Immediately after, we explored the effect of alcohol on the expression of DNMTs in tumor tissue and SiHa cells.” What does ‘immediately after’ refer to? Have further studies been conducted, the results of which are not included in this study? If so, it would be pertinent to include them, as the current results are insufficient to support the claims made.
- Overall, this discussion section serves more as a literature review (better suited for an introduction) and does not discuss in any detail the findings of this particular study. I would suggest a rewrite of the entire section with better emphasis on the purpose of the discussion section. The exception would be the latter part of this section, which contains some useful information to better clarify some findings herein – even so, these should be better worded to clarify the association between observation and HYPOTHESISED explanations (none of them being proven as such herein, as they are understandably beyond the scope of the study).
- The earlier part of this statement was not presented as data herein: “We found that alcohol can affect the occurrence of tumors by altering the level of intracellular DNA methylation.” What metric was used to quantify “occurrence of tumors”? Was this inferred from the methylation levels? If so, this is not an appropriate finding of the paper, and should be worded appropriately as an inference.
- Closing sentences of the section attempt to address some flaws in the study, but there are far more glaring issues regarding potential biases, presentation of results, and interpretation of data that should be addressed as limitations of the study. Arguably, without correction for these issues, some of which have been commented upon in this review, the study cannot warrant publication, and even with some corrective effort, they will need to be made explicit as limitations of the study.
- Fig. 8 provides a useful, concise representation of the hypothesised model explaining the study’s results. However, it is not referenced to at all in-text, and should be presented as a working model only, as the results are far from conclusive in supporting it. A better discussion section could be built around this figure with appropriate references to literature to better justify the model.
Methods
- The ethics grant / reference number should be provided here: “All experiments were approved by the Animal Research Ethics Committees of School of Public Health Jilin University.”
- Although it is mentioned that two-fold dilutions are used to prepare TMA and TLA solutions from TMA concentration at 3450mg/kg, the exact concentrations should be provided here to avoid misunderstanding. Additionally, the phrasing of the sentence outlining this process could be reworded for clarity. Since this method was outline, it begs the question why some results are presented with these categories, while others are presented with alcohol dosages of 200mM, 400mM, and 800mM? Some results, e.g. Fig.4, do not reflect any advantage of using the latter, especially when observable results are on the timescale of 4 – 6w, whereas the latter data is presented between 3 – 6d. It seems that there are two sets of experiments here (for mice survivability and for cell cytotoxicity), but there is no coherent reason for why the timeframes selected are so vastly different, and the experimental design here could be better presented to make clear the differences between the two sets of experiments (additionally, the difference in interpretation of results also warrants addressing in the discussion section).
- For the cell cytotoxicity set of experiments, it is mentioned that “(the final concentrations were 1600 mM, 800 mM, 400 mM, 200 mM, 100 mM, 50 mM, 25 mM, 12.5 mM, 6.25 mM and 0 mM).” However data is only presented for a subset of these – at 200mM, 400mM, and 800mM. It seems very untransparent to withhold the additional information, especially if further observation (or otherwise) of the purported trends can be observed or disproven from said data. In particular, the extension of dose-dependent data from the 800mM to 1600mM seems particularly potentially telling, and it would be worth including in the results section, regardless of observed statistical differences or otherwise (the latter being just as useful to deny any trend concluded from a limited dataset).
- Although ANOVA was used pairwise as mentioned “Differences between groups were compared by one-way ANOVA and pairwise comparison,” the general claim of trends observed in association to alcohol dosage increase would be better supported by a multivariate analysis, e.g. combining variables presented in Fig. 2. Have the authors attempted this, and if so, were the statistical findings still observed? If not, then it is highly suggestive that the findings are merely coincidental and not due to the suggested dose-dependence of alcohol concentration.
- Nowhere is it mentioned what the sample size is for the mice trials or any other experimental procedure, with one exception mentioned in parentheses below. This should be explicitly mentioned in the methods, abstract, and results section. Inferring from the data points in Fig.4, as well as the mention of “n=3” in Fig.6 (or is this specifically in reference to triplicates of homocysteine levels, mentioned as “The experiment was repeated 3 times, and the average value was taken.” If this is the case, then this is equally not intuitive, as Fig.6 is about mRNA expression levels of DMNT and Mecp) is it correct to conclude that only three mice were studied? If so, The statistical significance of the findings in Fig.4 are cast into much doubt. Both the statistical methodology and a power calculation to demonstrate that three mice are indeed enough to show such statistical significance is required here (perhaps in an appendix or even in the methods section).
References
- Number and quality of references seem rather lacking for such a novel claim. Better use of referencing is required to support many of the claims made herein, especially considering the tenuous nature of data and lack of rigorous supporting statistical evidence.
Author Response
Response to Reviewer 3 Comments
Abstract
- This statement in the abstract should be changed or removed: “However, the etiology and pathogenesis of cervical cancer are not very clear” as the pathogenesis of the vast majority of cervical cancer cases is well-linked to HPV infections. It may be true to say that those of unknown aetiology are poorly understood, but this sentence does not reflect this. Furthermore, this sets up the subsequent premise of the study, which is to investigate the effect of alcohol on methylation states in cervical cancer cells, perhaps better describing the latter subset (minority) of non-HPV cervical cancers.
Response: Thanks for your suggestion, we will change this sentence to‘Current research shows that persistent infection of high-risk HPV is the main cause of cervical cancer. However, not all cervical cancer is caused by HPV infection, which may also be related to other factors, such as nutritional status and lifestyle.’
- “In SiHa cells, the mRNA and protein levels of the DNMT family members and their receptors in the cells were significantly higher than those of the control group.” This statistical statement should be accompanied by the relevant threshold e.g. p<0.05.
Response: Thanks for your prompt, we have added threshold values (P <0.05) after statistically significant variables in the results section of the abstract.
- This statement seems very verbose and incorrect: “alcohol intervention can lead to the consumption of methyl-donor in cervical cancer cells and increase the expression levels of the DNA methyltransferase family members”, in particular regarding ‘alcohol intervention leading to consumption of …’ Perhaps paraphrasing would help better convey the conclusion regarding the relationship found between alcohol levels and expression levels of DNMT.
Response: Thanks for your suggestion, we have corrected the conclusion of abstract: ‘alcohol could influence DNMT expression by inducing methyl donor consumption, thereby causing cervical cancer cells to exhibit genome-wide hypomethylation’.
Introduction
- Similar to the comment in the abstract section, this statement doesn’t seem accurate: “the pathogenesis of cervical cancer is not very clear, and the main pathogenic factor of that is human papillomavirus (HPV) infection.” It may be true that we do not fully understand ALL the contributing factors to pathogenesis of cervical cancers, of which methylation states could be a major risk factor, but by far, HPV accounts for the highest risk amongst known factors, so it is disingenuous to suggest that the pathogenesis of cervical cancers, as a whole, is not very clear. As mentioned in the subsequent sentence, there are many other risk factors for the disease IN ADDITION to the main risk factor, HPV infection. To put this into context, perhaps mention of the relative risk contributions (percentagewise) would help readers.
Response: Thank you for your review. In the introduction, we deleted the statement that ‘the pathogenesis of cervical cancer is not very clear’. We described that the main pathogenic factor of cervical cancer is continuous infection of high-risk HPV.
- This statement, “the effect of DNA methyltransferase, a methyl group is added to the cytosine residues in a cytosine-guanine (CG) pair generating 5-methylcytosine, resulting in changes in DNA conformation, stability and interactions with proteins” merely addresses one type of DNA methylation, CpG methylation. Granted, this is the most common type of DNA methylation in humans, but as a general statement here, it is limited to only discuss CpG methylations, and incorrect to generalise that all methylation activity is of CpG methylations. Paraphrase here or include a more general statement about methylation. If the study here concentrates only on CpG methylation, it should be noted that this is the case, and why this choice was made.
Response: Thank you for your review. We are very sorry for the misunderstanding caused by the language expression. We have edited and polished the language of our manuscript and modified this part of expression, hoping to get a correct understanding.
- Inaccuracy in phrasing here, as ‘methyl’ is not a stand-alone substance, but a chemical side group (so it might be appropriate to paraphrase to ‘methyl-group’: “The main way for the body to obtain methyl is through food intake.”
Response: Thanks for your advice, we have corrected the statement of ‘methyl-group’ in our manuscript.
- Could the authors provide clarification for this statement: “In addition, studies have also reported that DNA methylation may directly supplement or replace HPV screening as a molecular diagnostic and prognostic marker for cervical intraepithelial neoplasia and cervical cancer[12,13]”, as the two references here do not indicate, or even suggest that methylation analysis could replace HPV screening, but does have the potential to supplement such screening (unless I am misreading them). If indeed, this is not what the references have mentioned, then removing this statement is essential as to not mislead the reader regarding the significance of findings here (of replacement, though the supplementary role of methylation screening can be kept).
Response: Thanks for your advice, we have deleted the incorrect description, and corrected it to ‘In addition, studies have also reported that DNA methylation may directly supplement HPV screening as a molecular diagnostic and prognostic marker for cervical intraepithelial neoplasia and cervical cancer.’
Results
- The first paragraph, including Fig.1 would be better suited in the methods section than in results section: “Mice with successful tumor bearing were given low, medium and high doses of alcohol solution by gavage, and the body weight and tumor tissue size were measured every 3 days. The tumor volume was calculated as follows: tumor volume (cm3 ) = length × width^2 /2. After gavage for 4w, half of the mice were sacrificed for follow-up experiments, and the other experimental animals continued to be treated with alcohol until 6w. The treatment process of tumor-bearing mice is shown in Figure 1.”
Response:: We accept your suggestion to adjust the introduction of the tumor-bearing mouse model in Figure 1 to the methods section.
- Fig.2 requires a lot more clarification. Legends and figure text should include the abbreviations used (TLA = low alcohol treatment, TMA = medium alcohol treatment, THA = high alcohol treatment. What is Tum?) For each of the panels, e.g. Fig.2A, are the apparent differences between groups statistically significant? If so this should be indicated in the figure with appropriate labels, e.g. an asterisk, and the appropriate p-value of significance mentioned. If they are not statistically significant, this should be made clear and the result interpretations modified accordingly. This statement does not belong in the figure legend, rather in the methods section: “At 4 weeks, half of mice in each group were sacrificed for subsequent detection, and the remaining experimental animals were treated with alcohol until to 6 weeks.” It is sufficient to mark the time periods of 4w and 6w sacrifice, with the addition to the figure of a legend explaining the abbreviations 4w and 6w. Similarly, for panel C, statistically significant results should be labelled. I assume the symbols for $ and #$ are used to signify some measure of statistical significance, but it is not made clear in the legend, so these results cannot be overtly interpreted. Again, the description for panel C belongs better in the methods section: “After the experimental animals were sacrificed, the weight of tumors was weighed”; instead, consider using a description of the RESULTS herein.
Response:: Thanks for your suggestion, we have revised the description of the notes and results in Figure 2 in detail, as shown in Manuscript 2.1.
- This sentence is not a result of the study and is better suited for the introduction or discussion section: “Methionine is one of the essential amino acids of the body, and it is also the precursor for the production of S-adenosylmethionine (SAM). SAM provides a carbon unit and is 4 converted into homocysteine via DNMT. SAM changes could cause the changing in homocysteine and DMNT.”
Response: Thank you for your review. Our intention was to briefly describe the effects of methionine, like folic acid and homocysteine, in one sentence and then describe the results. At the same time, we made a lot of revisions to the discussion section, and also added the content related to methyl donors (folic acid and methionine).
- This statement in-text does not match the data in Fig.3: “… folic acid levels in tumor-bearing mice treated with medium and high doses of alcohol decreased more significantly.” In Fig.3(A), only the THA group is marked as showing a significant difference between 4w and 6w, but the TMA groups isn’t marked as such. There may be a trend, as the authors suggest, but a dose-dependence cannot be established by comparison between two data points (either against the Tum sample, or between 4w and 6w); instead, a linear regression to show dose-dependence would be more appropriate. Similar argument for Fig.3(B) and (C). Furthermore, in panel C, the TMA data shows a significant increase in folate levels from 4w to 6w, which doesn’t corroborate the trend in other groups, as well as the progression from TLA to TMA to THA. This inconsistency puts into question the conclusion made (without context): “Meanwhile, the folic acid levels of tumor-bearing mice treated with high doses of alcohol in 6 w were more decreasing than 4 w (t=2.953, P=0.042)”, which is an untrue statement both statistically (no $ marker is given as per the legend), and the authors have conveniently omitted the statistically significant INCREASE in folate levels for the TMA group from 4w to 6w, of which, the 6w data is not significantly different from 0mM exposure to alcohol. Similar inconsistencies are observed in other panels, in particular, panel (I), where there is no apparent dose-dependent relationship between alcohol exposure and extracellular HCY. Indeed, it seems that the 6d levels of HCY are, where statistically demonstrable at 800mM alcohol, LOWER than at 3d, with a possible similar trend at other alcohol doses, though the latter are not statistically confirmed or denied. This observation warrants some addressing in the discussion section.
Response: Thank you for your review. We have revised the text description of Figure 3 in detail to make it clearer. At the same time, we have also made clear marks on the figure legend.
- Legend for Fig.3 is insufficient – all abbreviations should be elaborated, even if they seem to be apparent, e.g. Tum, HCY, TLA, TMA, THA etc.
Response: Thank you for your suggestion. In the process of modification, we have made a detailed description of the representation of each group in the method section, so that the display will be clearer and the figure legend will not be verbose. Also, abbreviations like ‘homocysteine (HCY) ‘are given in the introduction.
- “Current studies have shown that tumor tissues exhibit low levels of genome-wide methylation.” Be this as it may, this is not a generalised statement on the findings herein, and should be moved to the introduction section with appropriate referencing.
Response: Thanks for your advice, we did make a similar statement in the introduction, therefore, we have decided to remove this sentence from the results.
- Fig.4 seems to only provide observable data in panel A, and no statistical information (or even non-statistical trend) is observable in panel B – perhaps the latter is worth relegating to an appendix. Furthermore, the comprehensiveness of the presented data is questionable – see comments on the methods section below.
Response: Thanks for your prompt, we reanalyzed the data in Figure 4. We changed Figure 4 and the text description of DNA methylation levels in results part.
- Fig.5 has potential to demonstrate some qualitative data, however there is neither enough information in the figures and legends, nor is there sufficient explanation in-text. Some quantifiable measure of measuring the expression levels, with annotations on significant changes in the panels is required to prove conclusively any changes. Further to this, the authors should provide a description on sampling of these visual fields – how was it ensured that these are random samplings representative of a homogenous observation, instead of fields specifically chosen to demonstrate a perceived difference (observation bias)? The figure legend should include identification of immunolabelling – what markers are identified with blue / red staining?
Response: Thanks for your suggestion, we added quantitative data to better clarify the results of immunofluorescence, and we improve the description of visual fields acquisition.
- Overall, the text description of the results section is insufficient, and merely references the figures with the expectation that the reader will find them self-explanatory (which they are not, especially without comprehensive legends).
Response: Thanks for your suggestion, we have further modified the description of the result part, and marked the number of mice or the number of cell experiments repeated, so as to make the result more clear.
- Fig. 6 is an archetype of the above critique – it is not self-evident what the expected and observed trends are here, especially since the statistically significant changes are labelled, but the trends are not addressed. A more comprehensive in-text description on why there is a dose dependence of alcohol and observed expression levels of DMNT / Mecp mRNA but a reversal of the trend between 3 and 6 days is essential for interpretation of these figures.
Response: Thanks for your criticism, we have written a detailed description of what is shown in Figure 6.
- This statement is confusing: “After various cervical cancer cells were given different concentrations of alcohol…” Was more than one cervical cancer cell-line experimented upon (SiHa)? If so, the details of this were not provided. If not, this sentence requires paraphrasing. The continuation of the sentence is equally confusing: “… protein levels and trends of the DNMT family and its receptor were close to the mRNA levels and trends” does this mean that the trends were similar? Perhaps a better choice of words is required, because it is not apparent what values are ‘close’ between the two sets of experiments, and even if some values were, the different nature of the experiments and biological interpretations would render this interpretation meaningless. Again, there is lack of clarity in the explanation of results in-text, and the legend for Fig.7 is not very helpful in elucidating the situation. The mention of J software is made briefly in the figure legend, but neither interpretation nor methodology is given in the appropriate sections.
Response: Thanks for your review, we have revised these descriptions in our manuscript.
Discussion
- The introduction, results, and discussion sections should be made clear that the conclusions drawn are based on in-vitro cell line analysis, as well as mouse tumour hosting (and that these are not in-vivo samplings). Furthermore, sample size of each experiment should be clearly mentioned (simply n=3 in parentheses, or equivalent, where results are mentioned).
Response: Thanks for your prompt, we added the number of mice in each group or the number of cell experiment repeats in the result section, that is, ‘n=?’. This will make the results more clear.
- Neither references 14 nor 15 are contextually supportive of the statement in which they are referenced – they are not tumour methylation studies.
Response: Thank you for your review. We have corrected references 14 and 15, which focus on the relation of folic acid and DNA methylation. We removed the ‘tumor’ from the manuscript to make it more accurate.
- True as this statement may be, there is no logical coherence in placing this sentence where it is: “Meanwhile, some clinical studies have shown that low folic acid content (including blood folic acid concentration, folic acid intake, and/or folic acid intake) is associated with increased risk of cardiovascular diseases, multiple cancers, neural tube defects, and tumors [16]”.
Response: Thank you for your suggestions. We have made more modifications to the discussion part, hoping that it will be more logical, clear and easy to understand.
- This statement would serve better in an introduction section, as it does not discuss findings herein: “Alcohol consumption is also an important risk factor for tumor development, according to WHO data 3.5% of all cancer-related deaths are related to chronic alcohol drinking [19]. Alcohol could induce carcinogenesis in numerous organs, including the upper aero[1]digestive tract, liver, colorectum, and female breast, et al”. Further to this, correlation here (which are indeed risk factors) does not show causation, so this should not be used as a justification to validate correlations found in this study (which, themselves are not strongly statistically supported). The term ‘et al’ here is misused – perhaps the authors mean etc?
Response: Thanks for your advice, we have adjusted this description to the introduction section. The term ‘et al’ is indeed misused and has been changed to ’etc’.
- In this statement: “Immediately after, we explored the effect of alcohol on the expression of DNMTs in tumor tissue and SiHa cells.” What does ‘immediately after’ refer to? Have further studies been conducted, the results of which are not included in this study? If so, it would be pertinent to include them, as the current results are insufficient to support the claims made.
Response: Thank you for your review. It may be due to our poor choice of words that we did not do more experiments to clarify the point. Moreover, after treating tumor-bearing mice and SiHa cells with alcohol, we first detected the changes in the methyl donors, then the DNA methylation levels, and also observed the expression of DNMT family genes and proteins. ‘Immediately after’ is probably not the right word, so we would like to replace it to "next."
- Overall, this discussion section serves more as a literature review (better suited for an introduction) and does not discuss in any detail the findings of this particular study. I would suggest a rewrite of the entire section with better emphasis on the purpose of the discussion section. The exception would be the latter part of this section, which contains some useful information to better clarify some findings herein – even so, these should be better worded to clarify the association between observation and HYPOTHESISED explanations (none of them being proven as such herein, as they are understandably beyond the scope of the study).
Response: Thanks for your advice, we have made a lot of revisions to the discussion part, hoping that this can better explain the findings of this manuscript.
- The earlier part of this statement was not presented as data herein: “We found that alcohol can affect the occurrence of tumors by altering the level of intracellular DNA methylation.” What metric was used to quantify “occurrence of tumors”? Was this inferred from the methylation levels? If so, this is not an appropriate finding of the paper, and should be worded appropriately as an inference.
Response: Thank you for your reminding, and I'm very sorry for the misunderstanding caused by our improper wording. We tried to treat tumor-bearing mice with alcohol and observe the growth of tumors to reflect that alcohol can promote the growth of cervical cancer. Perhaps ‘growth of tumors’ is more appropriate than ‘occurrence of tumors’.
- Closing sentences of the section attempt to address some flaws in the study, but there are far more glaring issues regarding potential biases, presentation of results, and interpretation of data that should be addressed as limitations of the study. Arguably, without correction for these issues, some of which have been commented upon in this review, the study cannot warrant publication, and even with some corrective effort, they will need to be made explicit as limitations of the study.
Response: Thank you for your review. Indeed, at the end of the manuscript, we try to explain the limitations of this study. We understand that there are still some problems that cannot be clarified in this study. Based on the results of the current study, we would like to demonstrate that alcohol has an effect on cervical cancer, and which effects are manifested by changes in DNA methylation. Therefore, we have made a lot of modifications to the discussion section in order to better express our ideas.
- Fig. 8 provides a useful, concise representation of the hypothesised model explaining the study’s results. However, it is not referenced to at all in-text, and should be presented as a working model only, as the results are far from conclusive in supporting it. A better discussion section could be built around this figure with appropriate references to literature to better justify the model.
Response: Thank you for your advice. Figure 8 is indeed a model diagram for our summary of this article. We have made a lot of modifications to the discussion part and elaborated around the content in Figure 8, so as to make the discussion clearer.
Methods
- The ethics grant / reference number should be provided here: “All experiments were approved by the Animal Research Ethics Committees of School of Public Health Jilin University.”
Response: Thank you for your prompt, we have added the ethics grant number (No.2021-06-27) in the manuscript.
- Although it is mentioned that two-fold dilutions are used to prepare TMA and TLA solutions from TMA concentration at 3450mg/kg, the exact concentrations should be provided here to avoid misunderstanding. Additionally, the phrasing of the sentence outlining this process could be reworded for clarity. Since this method was outline, it begs the question why some results are presented with these categories, while others are presented with alcohol dosages of 200mM, 400mM, and 800mM? Some results, e.g. Fig.4, do not reflect any advantage of using the latter, especially when observable results are on the timescale of 4 – 6w, whereas the latter data is presented between 3 – 6d. It seems that there are two sets of experiments here (for mice survivability and for cell cytotoxicity), but there is no coherent reason for why the timeframes selected are so vastly different, and the experimental design here could be better presented to make clear the differences between the two sets of experiments (additionally, the difference in interpretation of results also warrants addressing in the discussion section).
Response: Thank you for your review. The specific dosages of alcohol tumor-bearing mice were given in methods part. SiHa cells were treated with the alcohol dosages of 200mM, 400mM, and 800mM, and the dose range was based on the experimental results of CCK-8. In this study, the alcohol treatment time of the tumor-bearing mouse model was 4w and 6w, respectively, while the alcohol treatment time of SiHa cells cultured in vitro was 3d and 6d, which were obtained according to the observation of the tumor growth state of experimental animals and the results of cell pre-experiment, and the time point of the two experiments was not inevitable. In the tumor-bearing mouse model, it takes a relatively long time to observe the effect of alcohol on tumor growth. However, in SiHa cells, due to the rapid proliferation in vitro culture, alcohol treatment for 3d and 6d is also a considerable time span, during which changes in the methyl donor and methylation state can be observed.
- For the cell cytotoxicity set of experiments, it is mentioned that “(the final concentrations were 1600 mM, 800 mM, 400 mM, 200 mM, 100 mM, 50 mM, 25 mM, 12.5 mM, 6.25 mM and 0 mM).” However data is only presented for a subset of these – at 200mM, 400mM, and 800mM. It seems very untransparent to withhold the additional information, especially if further observation (or otherwise) of the purported trends can be observed or disproven from said data. In particular, the extension of dose-dependent data from the 800mM to 1600mM seems particularly potentially telling, and it would be worth including in the results section, regardless of observed statistical differences or otherwise (the latter being just as useful to deny any trend concluded from a limited dataset).
Response: In the cytotoxicity experiment (CCK-8 method), we chose more doses (1600 mM, 800 mM, 400 mM, 200 mM, 100 mM, 50 mM, 25 mM, 12.5 mM, 6.25 mM and 0 mM), and described dose-time-response curves on the survival rate of SiHa cells, which treated with different doses of alcohol at different times, so as to reflect the toxicity of alcohol. However, in other subsequent experiments, an alcohol concentration close to the IC50 was selected as the medium dose, and upper and lower doses (0 mM, 200 mM, 400 mM and 800 mM) were selected as the experimental doses to observe the effects of alcohol on SiHa cell methyl donors and methylation status, while doses too low and too high were not selected.
- Although ANOVA was used pairwise as mentioned “Differences between groups were compared by one-way ANOVA and pairwise comparison,” the general claim of trends observed in association to alcohol dosage increase would be better supported by a multivariate analysis, e.g. combining variables presented in Fig. 2. Have the authors attempted this, and if so, were the statistical findings still observed? If not, then it is highly suggestive that the findings are merely coincidental and not due to the suggested dose-dependence of alcohol concentration.
Response: Thank you for your suggestion. In this study, one-way ANOVA and pairwise comparison were used for statistical analysis between each group. In particular, in Figure 2, statistical differences were analyzed and corresponding annotations were added to made the results more clear.
- Nowhere is it mentioned what the sample size is for the mice trials or any other experimental procedure, with one exception mentioned in parentheses below. This should be explicitly mentioned in the methods, abstract, and results section. Inferring from the data points in Fig.4, as well as the mention of “n=3” in Fig.6 (or is this specifically in reference to triplicates of homocysteine levels, mentioned as “The experiment was repeated 3 times, and the average value was taken.” If this is the case, then this is equally not intuitive, as Fig.6 is about mRNA expression levels of DMNT and Mecp) is it correct to conclude that only three mice were studied? If so, The statistical significance of the findings in Fig.4 are cast into much doubt. Both the statistical methodology and a power calculation to demonstrate that three mice are indeed enough to show such statistical significance is required here (perhaps in an appendix or even in the methods section).
Response: Thank you for your review. In the method of tumor-bearing mouse model, we have introduced how to group mice and the number of mice in each group. At the same time, when revising the manuscript results, we have also marked the number of mice used in each experiment or the number of cell experiments repeated by n=? representation.
References
- Number and quality of references seem rather lacking for such a novel claim. Better use of referencing is required to support many of the claims made herein, especially considering the tenuous nature of data and lack of rigorous supporting statistical evidence.
Response: Thanks for your review, we have added some references to support our argument in the process of revising the manuscript.
Round 2
Reviewer 2 Report
The authors have carried out experiments with three different cell lines and have not included the data in the current manuscript; in the current manuscript, they have used one cell line and concluded.
Reviewer 3 Report
Overall, the authors have addressed some of the previous iteration of comments, however, some major issues have been left unaddressed. Following are some major issues that, if left unaddressed, does not warrant yet publication. It is hoped that the authors may engage with the critiques and present the results in a manner suitable for publication.
- Figure texts are still not comprehensive as they do not include elaborations on abbreviations.
- This sentence is not a result of the study and is better suited for the introduction or discussion section: “Methionine is one of the essential amino acids of the body, and it is also the precursor for the production of S-adenosylmethionine (SAM). SAM provides a carbon unit and is 4 converted into homocysteine via DNMT. SAM changes could cause the changing in homocysteine and DMNT.” Although the authors have justified this comment in a rebuttal, stating their “intention was to briefly describe the effects of methionine, like folic acid and homocysteine, in one sentence and then describe the results,” I maintain that these sentences belong in the introduction section as they do not provide results from this study. This argument extends to most of the introductory sentences of the results section’s paragraphs: 1) “As an essential nutrient in the growth process, folic acid is closely related to DNA methylation. Folic acid acts as a one-carbon coenzyme in DNA synthesis and repair, and is directly involved in the tranfer of DNA methyl groups,” 2) “When methionine is consumed as a methyl donor, homocysteine is produced and released from the cell. Homocysteine is generated after the methyl donors are depleted, so detection of homocysteine levels can indirectly reflect DNA methylation levels.” Etc. (this is not an exhaustive list but the principle should be extended to all similar incidences. The major limiting issue with the current state of manuscript is the interpretation of results, coherence of argument, and a lacking discussion section.
- Section 2.6 and fig.7 still lacks adequate explanation – is it overtly interpretable that although alcohol influences the mRNA levels of the various methyltransferases, this does not result in any change in the protein expression levels (with some exceptions)? The quantitative data does not seem to support the qualitative data here – for example, numerically in panel B, there is no significant change in expression levels of DNMT1, but the 3d panel of A shows an increase in band intensity. Panel C shows a significant increase in DNMT-3a levels at higher alcohol doses, which is apparent in panel A but unmarked. Similarly, there is an apparent statistical increase for the 3d (but not 6d) expression for Mecp in panel E, which should me marked in panel A. What are the corresponding qualitative data for GAPDH, which is only present in panel A, but no graph given? This is useful data for the control.
- The legend for Fig.8 is non-descriptive and should be changed.
- The discussion section still lacks cohesion and the only major change made is the removal of a paragraph (that is better suited to the introduction section), which does not address the prior comment: Closing sentences of the section attempt to address some flaws in the study, but there are far more glaring issues regarding potential biases, presentation of results, and interpretation of data that should be addressed as limitations of the study. Arguably, without correction for these issues, some of which have been commented upon in this review, the study cannot warrant publication, and even with some corrective effort, they will need to be made explicit as limitations of the study.